# Food retail environments, extreme weather, and their overlap: Exploratory analysis and recommendations for U.S. food policy

**Benjamin Scharadin**[1]*, **Chad Zanocco**[2], **Jacqueline Chistolini**[3]

**1** Department of Economics, Colby College, Waterville, Maine, United States of America, **2** Department of Civil and Environmental Engineering, Stanford University, Stanford, California, United States of America, **3** Department of Statistics, Colby College, Waterville, Maine, United States of America

\* bpschara@colby.edu

## Abstract

Extreme weather events are increasing in frequency and severity due to climate change, yet many of their impacts on human populations are not well understood. We examine the relationship between prior extreme weather events and food environment characteristics. To do so, we conduct a U.S. county-level analysis that assesses the association between extreme weather events and two common food retail environment dimensions. Overall, we find a relationship between higher levels of historic extreme weather exposure and lower food availability and accessibility. In addition, we find heterogeneity in association across the distribution of the number of extreme weather events and event type. Specifically, we find that more localized extreme weather events are more associated with a reduction of access and availability than broad geographic events. Our findings suggest that as extreme weather events amplify in intensity and increase in frequency, new approaches for mitigating less acute and longer-term impacts are needed to address how extreme weather may interact with and reinforce existing disparities in food environment factors. Furthermore, our research argues that integrated approaches to improving vulnerable food retail environments will become an important component of extreme weather planning and should be a consideration in both disaster- and food-related policy.

## 1. Introduction

### 1.1 Background and motivation

Extreme weather events have exacted a devastating human toll in the past decade. In 2022 alone, Pakistan experienced the worst flooding in the country's history, killing thousands and leaving millions homeless [1]. Summer heatwaves in Europe broke records in the United Kingdom and France and were responsible for over 20,000 heat-related deaths [2]. In the United States, hurricanes caused tens of billions of dollars of damages with hundreds of associated fatalities. A historic North American winter storm generated intense snowfall and frigid temperatures across much of the U.S., subjecting one-third of the U.S. population to wind-chill alerts and left over 1.5 million households without power during its peak [3, 4]. While the

**Data Availability Statement:** There are three data sets used in the analysis of the manuscript. The Food Environment Atlas from the USDA's

Economic Research Service, county level demographic data from the U.S. Census, and the Spatial Hazards Events and Loss Database for the United States. The Food Environment Atlas is publicly available and downloadable (https://www.ers.usda.gov/data-products/food-environment-atlas/data-access-and-documentation-downloads/). The Census data is also publicly available and downloadable from the following URL (https://data.census.gov/mdat/#/). The variables and observations used in this study for these two data sets are available in the Supporting information files. Data for the SHELDUS data set are not included in the Supporting information files. The Spatial Hazards Events and Loss Database is a proprietary data set managed through the Center for Emergency Management and Homeland at Arizona State University. As a result, we are not able to provide a publicly available version of these variables. Readers can find more information and contact information at the CEMHS ASU website (https://cemhs.asu.edu/sheldus) to request access and pay the data acquisition fee.

**Funding:** The authors received no specific funding for this work.

**Competing interests:** The authors have declared that no competing interests exist.

scientific understanding of recent extreme weather events and human-caused climate change varies by event type, with extreme heat and high precipitation among the most attributable [5, 6], the severity and frequency of these events are only expected to increase as climate change progresses in the future.

Although these extreme weather events are highly destructive in terms of direct impacts on populations, it is also important to consider their other impacts in the context of on-going global challenges. There are high rates of food insecurity in both developed countries [7] and developing countries, where it more often takes on the form of undernutrition [8]. At the same time world obesity rates are trending upwards with areas in the Middle East, Central and Eastern Europe, and North America having among the highest levels [9]. The co-occurrence of these two events in place and time with the third pandemic of climate change increasing the frequency and severity of extreme weather events has been deemed a Global Syndemic [10]. The food system plays a critical role in each of these concurrent pandemics, both in terms of contributing factors and solutions, but research connecting the topics is limited in scope.

The focus of research connecting the food system with extreme weather events has focused on production and aspects of the supply chain (e.g., extreme heat, drought, crop damages, etc.) [11–15] with a recent study in Australia finding that fruit, vegetables, and livestock sectors are most likely to be impacted post-extreme events, with rural communities being more impacted [16]. Literature discussing consumer impacts focuses on post extreme event food-related concerns, covering a variety of topics including the role of social capital in event related food insecurity mitigation [17], emergency food reserves [18], access to food assistance networks (pantries) post event [19], food safety hazards associated with extreme weather event types [20], and even risk of food poisoning during power outage events [21]. While these are critical for post-event management and recovery, to date, there has been little discussion about post-event impacts across traditional food retail environment (FRE) dimensions, such as the availability of stores and households' ability to access them.

Therefore, an underexplored area of research is how extreme events intersect with, and (re) produce, existing food insecurity and other negative FRE phenomena, such as food deserts. This is particularly important as less acute food-related measures, such as longer-term impacts on food availability and access in the aftermath of events, can have lasting consequences for populations in the U.S. and elsewhere that extend far into the event recovery period and beyond. While some prior research has explored co-benefits of extreme weather and food-related policy planning [22], such studies have been event specific and have had a narrow focus. In contrast, we aim to consider the relationship between extreme event exposure and dimensions of the FRE to better understand how policy can both improve household access to healthy, nutritious foods, while simultaneously increasing the resilience to extreme weather events.

## 1.2 Research questions

Given these conceptualizations of food environments and extreme weather events, and our lack of understanding in how they may interact, we pose the following research question to broadly motivate our research inquiry: What, if any, is the relationship between the occurrence of extreme weather events and populations with limiting FRE? We further formalize this motivating research question with four related research questions, each closely linked to our analytical approach and data:

RQ1: What is the spatial relationship between extreme weather and food access?

RQ2: What is the quantitative relationship between extreme weather and food access?

RQ3: How does the relationship differ across the distribution of food access and availability?

RQ4: How does the relationship differ across storms with a local versus broad impact?

## 2. Materials and methods

As described in the previous research questions, we seek to understand the relationship between geographic areas with higher exposure to extreme weather events and areas that have lower access and availability to food retailers. Investigating this relationship requires multiple approaches to provide a more nuanced understanding of potentially important characteristics. Therefore, we first conduct a descriptive spatial analysis that visually compares extreme weather event locations and food access and availability measures (RQ1). Although this will provide important spatial context, this approach is not able to account for numerous confounding variables that may be present. Therefore, we conduct several regression analyses to establish a quantitative relationship and control for important related variables, described below.

### 2.1 Measure description, data sources, and variable construction

**Food retail environment measures.** Colloquially the FRE is often discussed with the term "food desert", however, fully characterizing the FRE requires more nuance. Caspi et al. [23] characterize the FRE with five dimensions, namely: availability, an adequate supply of healthy food; accessibility, the location of healthy food; affordability, the price and perceptions of price of food; acceptability, attitudes about the attributes of the FRE; and accommodation, how well the FRE adapts to local needs. Alternative, but still commonly cited characterizations include convenience and desirability [24], which make a distinction between the *community* nutritional environment and the *consumer* nutritional environment [25], with still others adding the organizational nutrition environment [26]. These dimensions aim to capture components of the "local" FRE by including farmers' markets and farms with direct food sales [27].

Although past literature has considered a wide range of FRE dimensions, availability and accessibility are the two foundational dimensions considered most often [23]. This is partially due to readily available geographic data on store location and increasing advancements in Geographic Information Systems (GIS) [28], but also denotes that a store must first exist in a location and that location must be reasonably accessible by households in the geographic area before other dimensions can be constraints. In addition, these two dimensions relate to the community nutritional environment, i.e. the built environment describing store location and density, rather than the consumer nutritional environment, i.e. how individuals utilize features of the FRE [25]. Therefore, we focus our analysis on measures of availability and accessibility, two common aspects of food insecurity, because our goal is to understand the relationship between exposure to extreme weather events and the built FRE in areas where they frequently occur.

Common availability measures include store counts and store density measures within an administrative boundary [29–31]. Although human activity is not bound by administrative boundaries (e.g. county lines), data collection within these units is practical. We continue with this common practice by using a density measure (i.e. store count divided by the population) to control for population density. Store venue types may be impacted by extreme weather events differently based on the resilience of their supply chain, variety of products, and size. Therefore, we consider the density of grocery stores, where a majority of food at home purchases are made [32], convenience stores, which have high retail availability [33], and supercenters, where shoppers may more readily practice bulk buying [34]. Accessibility is

commonly measured by the proportion of the population that is low-income, has low-access to grocery stores, and does not have access to a vehicle [35]. The United States Department of Agriculture (USDA), who maintains the Food Environment Atlas (FEA), considers a household to be low-income if their annual family income is at or below 200 percent of the Federal poverty threshold for their family size and a household to be low access if they live farther than 1 mile from the nearest supermarket, supercenter, or large grocery store for an urban area or farther than 10 miles for a rural area [36]. Therefore, we consider the percent of the population that is considered low-income, low access, and does not have access to a car as our measure of accessibility.

We utilize the USDA Economic Research Service's FEA, a publicly available dataset with predominantly county level data [36] for our FRE accessibility and availability measures. The full dataset includes information on numerous aspects of the FRE including access to food at home and food away from home, retailers, food insecurity assistance program participation rates, prevalence of local farms and direct to consumer sales, and various socioeconomic and demographic information. We use four county level measures; the number of grocery stores per 100,000 people, supercenters per 100,000 people, and convenience stores per 100,000 people as availability measures and the percent of a county defined as low access and low income without access to a vehicle as the accessibility measure. There are the four measures included in $FRE_{ct}$ in Eqs (1) and (2). For race and ethnicity controls, $Race_{ct}$, we use the percent of the population identifying as non-Hispanic Black, the percent of population identifying as Hispanic, the percent of population identifying as Asian, and the percent of the population identifying as Asian American and Pacific Islander (AAPI). The FEA indicates counties as either metropolitan or rural following the Office of Management and Budget's definition. Counties are considered metropolitan if they are in Metropolitan Statistical Area (MSA) containing a core urban area of 50,000 or more population. All counties that are not part of a MSA are considered rural.

The FEA is updated frequently and includes two distinct years at a 5-year interval. First round observations for our variables of interest are from 2010 and 2011 and second round observations are from 2015 and 2016. Some race and ethnicity controls are not available for both years. In this case, data is supplemented using the American Community Survey, which is one of many sources of data used to compile the FEA. Past literature has used the FEA to investigate the relationship between the FRE and obesity rates [37, 38], how these relate to disparities by race [39], and if access to direct to sale retailers can mitigate the impact [40]. Table 1 presents descriptive statistics at the U.S. county level for the variables of interest from the FEA.

**Table 1. Descriptive statistics for variables of interest from the Food Environment Atlas per U.S. counties.**

| Variable Description | Mean | Standard Error |
|---|---|---|
| The number of grocery stores per 100,000 people | 24.61 | 0.27 |
| The number of supercenter per 100,000 people | 1.73 | 0.03 |
| The number of convenience stores per 100,000 people | 59.51 | 0.39 |
| Percent of the population that is low-income, low access with no vehicle. | 3.19 | 0.04 |
| Percent population identifying as non-Hispanic Black | 4.61 | 0.14 |
| Percent population identifying as Hispanic | 4.24 | 0.13 |
| Percent population identifying as Asian | 0.59 | 0.02 |
| Percent population identifying as AAPI | 0.97 | 0.07 |
| Metropolitan County (1 = Yes, 0 = No) | 0.38 | 0.01 |

**Extreme weather events.**    As extreme events become more frequent and severe, so does the increasing potential for multiple exposures to connected extreme events [41]. Under this backdrop of increasing risks posed by extreme weather events, mounting evidence suggests that the impacts of and exposure to events caused by climate change are being experienced unequally across human populations–across countries, regions, cities, and even neighbor-hoods within the same city [42–45]. Such unequal exposure and impacts are typified by the distribution of extreme events and their impacts in the U.S. [46, 47]. In the U.S., exposure to some extreme events are more geographically bounded by climate zones and ecosystems, such as Atlantic hurricanes impacting the gulf coast and eastern seaborn and wildfires in the west. However, other events, such as extreme heat, cold, or extreme precipitation/flooding are not as geographically bounded and could potentially impact nearly every area of the U.S. [48].

Measurement of extreme events presents its own challenges, such as what threshold consti-tutes an extreme event based on extant criteria [49]. These criteria can be probability-based, such as if the magnitude of the event exceeds a chance of occurrence (e.g., <5%) given a refer-ence period (e.g., 1960–2000), or impacts-based, such as quantifying heat waves using the number of consecutive days over 100˚F [50]. The latter often relies on exceeding some impact threshold (e.g. a certain level of damages/costs). However, choosing the appropriate threshold is difficult because quantifying the impact of extreme events on heterogeneous populations presents additional challenges. Reporting practices for common measures, such as property damage, injuries, and fatalities, can under-reported for certain populations and may not well-characterize the full extent of certain event types [51]. Moreover, exposure to extreme events does not result in the same distribution of impacts, even for populations located within the same geographic area. Differences in housing types, access to resources, vulnerability, and other characteristics contribute to such heterogeneity. For example, recent literature found that those who are living in poverty and those who are racial/ethnic minorities self-report higher levels of impacts from extreme weather events [47]. Our research aims to investigate if one potential reason for these higher self-reported impacts from similar exposure levels is because racial/ethnic minorities often reside in FRE that have limited access and availability to food retailers.

Prior research about extreme weather event impacts on populations broadly falls within three research scopes: (1) consideration of the impacts of a single extreme event (e.g., [52]; (2) consideration of the same event type across space and time (e.g., [53]); (3) consideration of multiple event types across space and time (e.g., [54]). In this research, we focus on exposure to extreme weather events measured by historical count of events, their event type, and their location, rather than impact measures because, as mentioned above, common impact mea-sures may not be well suited to accurately characterize impacts across a range of populations. In addition, impact measures, such as property damage, will be closely related to the quality of the FRE because higher median income and house values will be positively correlated with greater access and property damage estimates. This association makes it difficult to disentangle the relationship to impact measures from latent variables like metropolitan or rural status. There are multiple sources of data on extreme events at population scales that can be utilized to implement this type of approach, including the National Storm Database, NOAA's Billion-Dollar Weather and Climate Disasters dataset, and the Spatial Hazards Events and Loss Data-base (SHELDUS). For this application, SHELDUS is the preferred data set because it offers spatial hazard level data, by discrete event, at a geographic and temporal resolution that aligns with available FRE availability and accessibility measures. This allows us to apply a measure of extreme weather event exposure (event type occurrence within a geography) in our analysis.

We utilize the SHELDUS dataset [55], a county level dataset that provides information across 18 different hazard types, for our data on extreme weather exposure. From the 18 event

**Table 2. Descriptive statistics for variables of interest from the Spatial Hazards Events and Loss Database per U. S. counties.**

| Variable Description | Mean | Standard Error |
|---|---|---|
| Count of extreme weather events | 14.87 | 0.16 |
| Count of tornados | 0.98 | 0.02 |
| Count of severe thunderstorms | 6.55 | 0.08 |
| Count of wind events | 2.30 | 0.05 |
| Count of flood events | 2.50 | 0.04 |
| Count of winter weather events | 1.11 | 0.03 |
| Count of hurricane or tropical storms | 0.19 | 0.01 |
| Count of wildfires | 0.22 | 0.01 |

types collected, we select 11. These include hurricane/tropical storm, wildfire, tornado, winter weather, severe/thunder, lightning, coastal, flooding, heat and wind. Natural disasters, e.g. avalanches, and low impact events, e.g. fog and hail, are excluded because there is a low probability of these types of events interacting with the FRE. SHELDUS is commonly used in literature investigating the spatial patterns of extreme weather events and their associated losses across the U.S. [56–60].

Complex events can be attributed to multiple hazard types, with each hazard type counted as a separate event within the SHELDUS database [60]. In order to avoid double counting, events are assigned a primary event based on a hierarchy of hazard types. Larger events, such as hurricanes, are placed at the top and general events that are commonly associated with other events, such as severe wind, are given lower priority. For example, a tornado event that involved lightning and high wind would only be considered one tornado event, instead of a total of three events; one tornado, one lightning, and one high wind.

Reporting in extreme weather events can experience large year to year variation both because of event count variation and an increase in event reporting over time through advances in loss reporting procedures [60]. This has the ability to introduce measurement error into the spatial map comparison. In order to mitigate the impact of this year to year variation, we aggregate event counts into a five-year range from 2013–2017. This range centers on the most recent year available for the FEA. We limit the extreme weather map to the number of extreme weather events for high property damage counties to more accurately capture exposure to events that may cause disruptions. To minimize the impact of large variation in property values across the U.S., we standardized the SHELDUS property damage values by using median house value per county in the American Community Survey. Once standardized, only events with damage values above the median standardized damage value are used in the spatial map comparison. In contrast, we do not aggregate or filter the data in the regression analysis because we can directly control for each of these issues with the regression specification. Table 2 presents descriptive statistics for variables of interest from SHELDUS.

## 2.2 Regression analysis

We utilize various forms of regression analysis to address RQ2 through RQ4. A spatial map comparison can highlight potential visual patterns, but is limited because it is not able to control for multiple confounding variables at once. For example, there is more likely to be a higher density of grocery stores in metropolitan areas and also more likely to be higher reporting rates of extreme weather events given that a portion of the SHELDUS data is sourced through insurance claims. Therefore, to avoid drawing conclusions based on unaccounted for latent

variables, we estimate the following equation.

$$\boldsymbol{FRE}_{ct} = Events_{ct} + \boldsymbol{Race}_{ct} + Metro_{ct} + \boldsymbol{Region}_c + Year_t + \varepsilon_{ct} \qquad (1)$$

$\boldsymbol{FRE}_{ct}$ represents a vector of the four food retail environment measures; grocery stores per 100,000 people, supercenters per 100,000 people, convenience stores per 100,000 people, and the percent of the population that is both low-income and low access without access to a vehicle in county c and year t. $Events_{ct}$ represents the number of extreme weather events that occurred in county c and year t. $\boldsymbol{Race}_{ct}$ represents a vector of the proportion of the population in county c that identifies as a particular race or ethnicity. Controlling for race and ethnicity is important because many built environments in the U.S., particularly food retail environments, are highly correlated with race and ethnicity because of historical and institutionalized policies [61]. Finally, $Metro_{ct}$ controls for if county c is considered metropolitan, Region is a fixed effect controlling for the USDA region of county c, and Year is a time fixed effect controlling for the year.

In order to address RQ2, we estimate Eq 1) using three specifications that build towards a preferred approach. First, we estimate a baseline specification using Ordinary Least Squares to understand how the alterations in future specifications improve the estimation. The baseline matches Eq 1) exactly, which results in assuming that each additional extreme weather event relates to the FRE outcome variable linearly. In the baseline specification, this means the additional impact is independent of the number of weather events. However, the impact of an additional weather event on availability and accessibility may compound the effect of previous weather events or have less of an impact than the previous event [62]. In the second specification, we include a term for the number of extreme weather events squared to account for this possible non-linearity. Finally, in the third specification, we estimate a log-linear regression model. An additional extreme weather event may not be linearly associated with the count of food retail venues, but rather linearly associated with a percent change in food retail venues because of the large variation in store density across counties. The log-linear specification will address this concern and prevent the large number of counties with no supercenters from biasing the estimation.

To address RQ3, we continue with log-linear form of Eq 1) and use quantile regression separately for each FRE outcome variable to investigate how the relationship between extreme weather events and food access and availability may differ across the distribution of extreme weather events. Quantile regression is particularly useful when there is reason to believe that the relationship between an independent variable and dependent variable may differ across the distribution of the dependent variable and has been used in both FRE literature [63, 64] and extreme weather literature [65, 66]. In this case, it may be possible that areas with relatively high food access may be more impacted by extreme weather than areas with relatively low food access because there are more stores to be harmed by the events. Alternatively, a less robust FRE may be more disrupted by even a small number of extreme weather events because the overall system is more fragile. Therefore, there is no expected a priori relationship. Understanding these differences, or lack thereof, can help guide policy around extreme weather response and highlight additional positive outcomes from policies aimed at fortifying low food access and availability areas independent of extreme weather events.

Finally, we address RQ4 by disaggregating the extreme weather events by type. We first group events into more localized events, such as tornadoes and severe thunderstorms, and broad geographic events, such as winter weather storms and hurricanes. These types of events may have different associations with FRE measures because of their unique characteristics. Broad geographic events may cripple supply chains over a much larger region, but are also

often forecasted with advance warning. In contrast, more localized events may only hinder a small portion of the FRE supply chain, but it can be harder to forecast their timing. Similar to understanding differences across the distribution of the number of extreme weather events, understanding how specific weather event types interact with food access and availability can better inform policy response and preparedness. For example, if more localized events, such as tornadoes, are associated to a greater extent with lower food access and availability, policy makers can target geographic areas that are at risk for both.

$$FRE_{ct} = EventType_{ct} + Race_{ct} + Metro_{ct} + Region_c + Year_t + \varepsilon_{ct} \qquad (2)$$

Eq 2 provides the modified specification to estimate the log-linear specification using counts by event type instead of the overall count of events. Each variable is defined similarly as Eq 1), with $EventType_{ct}$ representing a vector of the count of events by type. We estimate Eq 2) separately for more localized events and broad geographic events.

## 3. Results

The following section presents the results corresponding to each of our four research questions. We first present a spatial map comparison and discuss both similarities and differences across regions of the U.S. (RQ1). We then present three tables that correspond to RQ2 through RQ4. We will first present and interpret the results for the estimation of Eq 1) building to the log-linear model. Next, we address RQ3 by presenting and discussing the results for the quantile regression, which allow for comparison across the distribution of the number of extreme weather events. Finally, we present the results for the estimation of Eq 2), i.e. separate regressions for localized and geographically broad extreme weather events.

### 3.1 RQ1—What is the spatial relationship between extreme weather and food access?

Fig 1 presents the spatial map comparison. Overall, visual similarities are greatest between extreme weather and access measures, rather than availability measures. The southeastern and southwestern United States, including areas of Mississippi, Alabama, Louisiana, Arizona, New Mexico and Western Texas as well as Northern Vermont and New York are all areas with high numbers of high property damage extreme events with higher percentages of the population defined as low income and low access with no vehicle. In Northern Vermont and New York, the extreme events are likely to be winter weather events, where low access to grocery stores and no vehicle could be a significant barrier to adequate resources during a major event. In the southern states, such as Alabama and Louisiana, the weather event is more likely to be a hurricane or tropical storm. Despite the storms being meteorologically different, their impact, e.g. loss of power, impassable roads, etc. are similar and especially debilitating for low-income and low access households without a vehicle.

There are also areas with high extreme weather counts and low grocery and convenience store availability, but the concentration for each store type differs. In general, grocery store availability is sporadically distributed across the U.S., but has particularly low density in a few areas with high extreme weather exposure, i.e. the southern California, eastern Arizona, and the Texas gulf coast. Convenience store density is a bit more concentrated in the South and has low availability in high exposure areas, such as Florida and the Lake Erie region. Although convenience stores traditionally have a more sparse selection of healthy food, they are still an important source of food and beverages during an extreme weather event. In summary, these maps serve to highlight visual relationships; however, these patterns are subject to numerous confounding variables due to socioeconomic factors that are associated with living in low

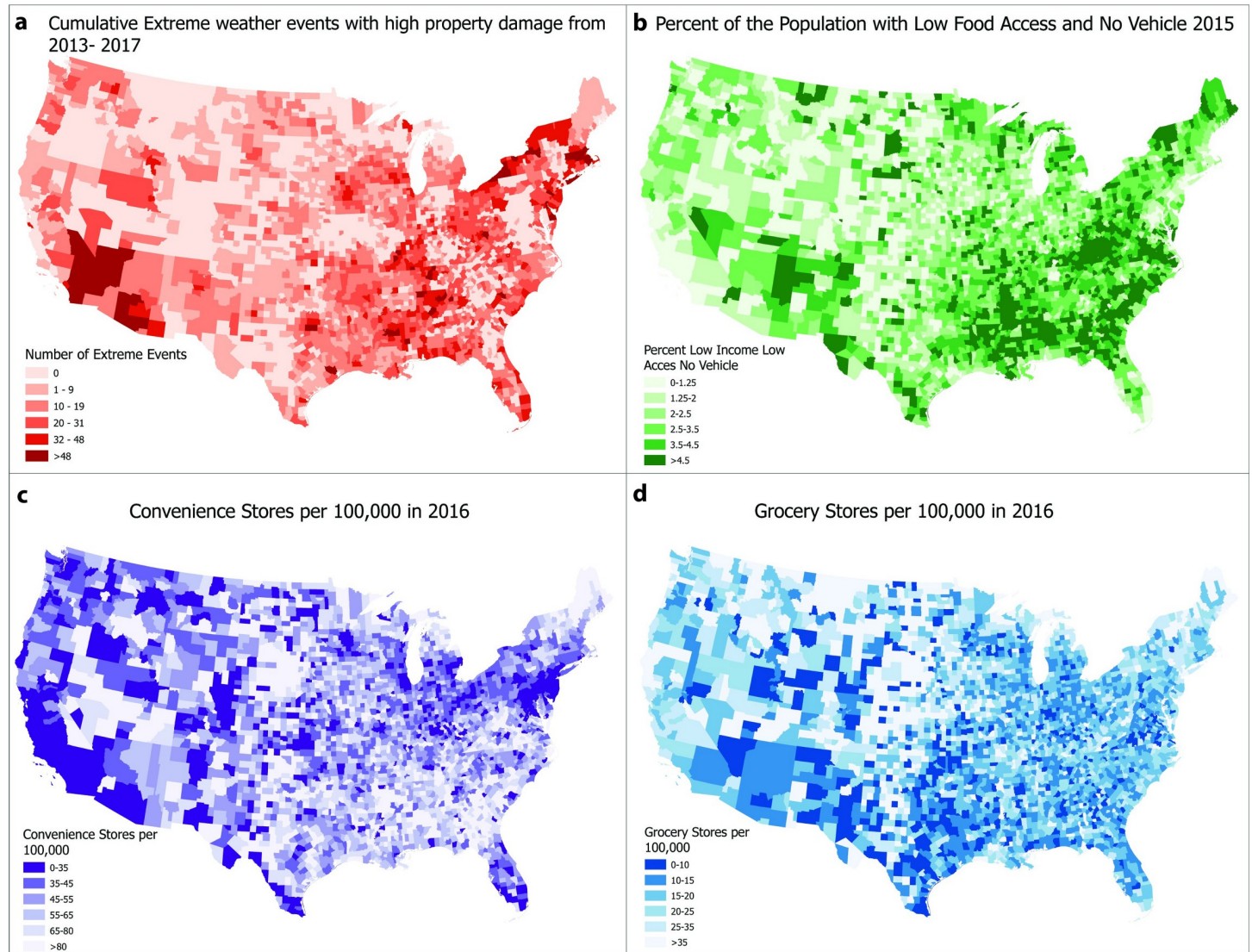

**Fig 1. Four-panel map of U.S. counties showing (a) high property damage extreme weather events, (b) the proportion of low-income low access households with no vehicle, and (c) convenience and (d) grocery store densities.** Data for extreme events comes from SHELDUS, and food access and grocery and convenience store location data comes from the FEA. For the cumulative extreme events and access maps, lighter colors represent lower access and lower numbers of extreme events. Maps representing availability measures use a reversed color scheme where lighter colors are associated with larger numbers of grocery and convenience stores.

access and disaster prone areas [47]. Therefore, they motivate the use of regression analysis in which these factors can be incorporated.

### 3.2 RQ2—What is the quantitative relationship between extreme weather and food access?

Table 3 presents the results for three specifications of Eq 1) that address RQ2. The first four columns correspond to the baseline specification, i.e. matching Eq 1) exactly; one for each food environment outcome measure. The second four columns present a similar specification, but include the number of events squared. The final four columns correspond to the log-linear specification. All specifications include USDA region fixed effects to control for broad

**Table 3. Estimation results for three specification of Eq 1.**

| VARIABLES | Baseline Specification | | | | Weather Events Squared | | | | Log-Linear Specification | | | |
|---|---|---|---|---|---|---|---|---|---|---|---|---|
| | Grocery | Super Center | Conv. Store | Low-Access | Grocery | Super Center | Conv. Store | Low-Access | Grocery | Super Center | Conv. Store | Low-Access |
| Number of Events | -0.206*** | 0.017*** | -0.259*** | -0.0029 | -0.5365*** | 0.0478*** | -0.6964*** | 0.0151** | -0.0153*** | -0.0120*** | -0.0101*** | 0.0074*** |
| | (0.023) | (0.003) | (0.032) | (0.003) | (0.056) | (0.006) | (0.078) | (0.008) | (0.002) | (0.002) | (0.001) | (0.002) |
| Events Sqaured | | | | | 0.0067*** | -0.006*** | 0.0089*** | -0.004*** | 0.0002*** | 0.0001*** | 0.0001*** | -0.0001*** |
| | | | | | (0.001) | (0.000) | (0.001) | (0.000) | 0.000 | 0.000 | 0.000 | 0.000 |
| Percent NH Black | 0.0302 | -0.0073** | 0.214*** | 0.0467*** | 0.0312 | -0.0074** | 0.2150*** | 0.0467*** | 0.0029*** | -0.0044*** | 0.0031*** | 0.0092*** |
| | (0.027) | (0.003) | (0.038) | (0.004) | (0.027) | (0.003) | (0.038) | (0.004) | (0.001) | (0.001) | (0.001) | (0.001) |
| Percent Hispanic | -0.107*** | -0.0025 | -0.053 | -0.0213*** | -0.1175*** | -0.0012 | -0.0713* | -0.0205*** | -0.0037*** | -0.0065*** | -0.0022*** | -0.0084*** |
| | (0.029) | (0.003) | (0.040) | (0.004) | (0.029) | (0.003) | (0.040) | (0.004) | (0.001) | (0.001) | (0.001) | (0.001) |
| Percent Asian | 0.148 | 0.011 | -1.543*** | -0.0760*** | 0.1404 | 0.0114 | -1.5496*** | -0.0756*** | 0.0062 | -0.0579*** | -0.0358*** | -0.0507*** |
| | (0.152) | (0.016) | (0.212) | (0.021) | (0.152) | (0.016) | (0.211) | (0.021) | (0.004) | (0.004) | (0.003) | (0.005) |
| Percent AAPI | 0.275*** | -0.016*** | -0.091 | 0.2027*** | 0.2706*** | -0.0162*** | -0.0966 | 0.2030*** | 0.0062*** | -0.004 | 0.0007 | 0.0246*** |
| | (0.049) | (0.005) | (0.068) | (0.007) | (0.049) | (0.005) | (0.068) | (0.007) | (0.002) | (0.003) | (0.001) | (0.002) |
| Metro County | -9.125*** | -0.092 | -17.71*** | -1.168*** | -8.8978*** | -0.1123* | -17.418*** | -1.1811*** | -0.3364*** | -0.4117*** | -0.3060*** | -0.3542*** |
| | (0.567) | (0.061) | (0.787) | (0.076) | (0.566) | (0.061) | (0.786) | (0.076) | (0.015) | (0.019) | (0.011) | (0.019) |
| Constant | 32.32*** | 1.45*** | 71.35*** | 3.108*** | 34.8789*** | 1.2175*** | 74.7399*** | 2.9704*** | 3.3463*** | 1.4916*** | 4.2640*** | 0.9633*** |
| | (0.651) | (0.070) | (0.904) | (0.088) | (0.761) | (0.082) | (1.058) | (0.103) | (0.021) | (0.030) | (0.015) | (0.025) |
| Region FE | Yes | Yes | Yes | Yes | Yes | Yes | Yes | Yes | Yes | Yes | Yes | Yes |
| Year FE | Yes | Yes | Yes | Yes | Yes | Yes | Yes | Yes | Yes | Yes | Yes | Yes |
| Obs. | 6,002 | 6,002 | 6,002 | 6,000 | 6,002 | 6,002 | 6,002 | 6,000 | 5,877 | 3,392 | 5,935 | 5,974 |
| R-squared | 0.123 | 0.024 | 0.163 | 0.235 | 0.129 | 0.029 | 0.168 | 0.236 | 0.178 | 0.362 | 0.264 | 0.235 |

Standard errors in parentheses

*** $p < 0.01$,

** $p < 0.05$,

* $p < 0.1$

socioeconomic and cultural effects and year fixed effects to control for broad macroeconomic and climate trends that occur in a particular year. These results are not reported because the coefficient sign and magnitude are not easily interpretable in a real world context. The sign and significance of the control variables match expectations given past literature and generally stay consistent across specifications. Namely, a larger proportion of the population identifying as non-Hispanic Black is associated with lower access to supercenters, but higher access to convenience stores, while a county being non-metropolitan is associated with lower store density and a lower proportion of the population being low-income and low access with no vehicle.

In the baseline specification, the number of extreme weather events is significantly associated with store density for each store type, but is not significantly associated with accessibility. An additional extreme weather event is associated with 0.2 and 0.26 fewer grocery stores and convenience stores per 100,000 people, respectively. In contrast, the baseline specification suggests that an additional extreme weather event is associated with more supercenters per 100,000. In general, there are few differences in sign and significance when the number of extreme weather events squared is added into the model. In this specification, an additional extreme weather event is still associated with fewer grocery stores and convenience stores per 100,000 people, respectively, while an additional extreme weather event is again associated with more supercenters per 100,000. However, the significance of the square term indicates

the model accounts for important non-linearities in the number of extreme weather events. For example, one additional extreme weather event is associated with 0.53 fewer grocery stores per 100,000 people, but the impact of each additional event reduces the associated reduction.

In each of these first models, the result for supercenters, i.e. more extreme weather events associated with more supercenters, is counter-intuitive. However, this result may be biased because of the large number of counties with no supercenters. The final four columns in Table 3 present the results for the log-linear specification, which assumes the relationship is linear with respect to the percent change in food retail venues and excludes counties with no supercenters. The sign and significance for grocery stores, convenience stores, and low-income low access are consistent with the previous model. An additional extreme weather event is associated with a 1.53 percent fewer grocery stores, 1.01 percent fewer convenience stores, and a .74 percent increase in the proportion of the population that is low-income and low access. In contrast, an additional extreme weather event is now associated with a 1.20 percent reduction in the number of supercenters. This preferred specification suggests that counties with higher rates of extreme weather events on average have lower access and availability to food retail outlets.

### 3.3 RQ3—How does the relationship differ across the distribution of food access and availability?

In addition to understanding the average association of extreme weather exposure on food access and availability, it is useful for policy makers to understand how the relationship differs for counties with the most and least robust food systems. Therefore, Table 4 presents the results for the log-linear specification estimated using quantile regression. Similar to previous estimations, all specifications include USDA region fixed effects and year fixed effects. Furthermore, the sign and significance of the control variables again match expectations and generally stay consistent across quartiles.

In general, an additional extreme weather event is associated with a larger decrease in store density as the availability of stores increases. The increase in association is most pronounced for grocery and convenience stores. An additional extreme weather event is associated with a 0.49 percent decrease in grocery store density at the 25th percentile, while it is associated with a 2.01 percent decrease at the 75th percentile. Similarly, an additional extreme weather event is associated with a 0.47 percent decrease in convenience store density at the 25th percentile, while it is associated with a 1.49 percent decrease at the 75th percentile. One explanation for this result may be that counties in the highest quartile, those with a relatively high number of grocery and convenience stores per 100,000 people, have enough stores to react to extreme weather events, while counties in the lowest quartile do not. The lower magnitude association in the lowest quartile does not mean counties are well prepared for extreme weather events, but motivates different approaches from a policy perspective. These results may suggest different approaches to extreme weather impacts in high and low access counties.

The number of extreme weather events is only significant for the lowest quartile when considering accessibility. An additional event is associated with a 0.84 percent increase in the proportion of the county population that is low-income, low access, and does not own a vehicle. This relationship is near zero and insignificant at the median and the 75th percentile. Individual and household mobility may explain the difference across the distribution. A household with a higher income and car ownership is in a better position to relocate from a county that experiences a high number of extreme weather events or avoid moving into the county in the first place. In contrast, extreme weather events may lower property values and thus be attractive to lower income households that have less mobility to leave. This suggests that

**Table 4. Estimation results for a quantile regression on the relationship between events and access and availability.**

| Percentile | Grocery | | | Supercenter | | | Convenience | | | Low-Access | | |
|---|---|---|---|---|---|---|---|---|---|---|---|---|
| | 25th | 50th | 75th | 25th | 50th | 75th | 25th | 50th | 75th | 25th | 50th | 75th |
| Number of Events | -0.0049*** | -0.0123*** | -0.0201*** | -0.0085*** | -0.0111*** | -0.0110*** | -0.0047*** | -0.0102*** | -0.0149*** | 0.0084*** | 0.0007 | 0.0025 |
| | (0.002) | (0.002) | (0.002) | (0.003) | (0.002) | (0.003) | (0.001) | (0.001) | (0.001) | (0.002) | (0.001) | (0.002) |
| Events Sqaured | 0.0001** | 0.0002*** | 0.0003*** | 0 | 0.0001* | 0.0001** | 0.0001*** | 0.0001*** | 0.0002*** | -0.0002*** | -0.0001*** | -0.0001*** |
| | 0.000 | 0.000 | 0.000 | 0.000 | 0.000 | 0.000 | 0.000 | 0.000 | 0.000 | 0.000 | 0.000 | 0.000 |
| Percent NH Black | 0.0040*** | 0.0029*** | 0.0032*** | -0.0032** | -0.0027* | -0.0034*** | 0.0038*** | 0.0032*** | 0.0029*** | 0.0108*** | 0.0109*** | 0.0090*** |
| | (0.001) | (0.001) | (0.001) | (0.001) | (0.002) | (0.001) | (0.001) | (0.001) | (0.000) | (0.001) | (0.001) | (0.001) |
| Percent Hispanic | -0.0027*** | -0.0024*** | -0.0030*** | -0.0057*** | -0.0057*** | -0.0064*** | -0.0016** | -0.0024*** | -0.0032*** | -0.0104*** | -0.0066*** | -0.0051*** |
| | (0.001) | (0.001) | (0.001) | (0.002) | (0.001) | (0.001) | (0.001) | (0.001) | (0.001) | (0.002) | (0.001) | (0.001) |
| Percent Asian | 0.0099*** | 0.0036 | 0.0028 | -0.0811*** | -0.0653*** | -0.0521*** | -0.0394*** | -0.0376*** | -0.0380*** | -0.0957*** | -0.0619*** | -0.0515*** |
| | (0.003) | (0.003) | (0.004) | (0.009) | (0.010) | (0.009) | (0.008) | (0.006) | (0.006) | (0.010) | (0.009) | (0.009) |
| Percent AAPI | 0.0016 | 0.0045*** | 0.0088** | -0.0065*** | -0.0056 | -0.0061** | 0.0001 | 0.0002 | 0.0033* | 0.0173*** | 0.0205*** | 0.0217*** |
| | (0.003) | (0.002) | (0.004) | (0.003) | (0.004) | (0.003) | (0.002) | (0.003) | (0.002) | (0.003) | (0.003) | (0.007) |
| Metro County | -0.2536*** | -0.3225*** | -0.4025*** | -0.3062*** | -0.3590*** | -0.4356*** | -0.2706*** | -0.2932*** | -0.3397*** | -0.3044*** | -0.3226*** | -0.3485*** |
| | (0.021) | (0.015) | (0.022) | (0.027) | (0.022) | (0.022) | (0.014) | (0.010) | (0.010) | (0.019) | (0.019) | (0.015) |
| Constant | 2.8954*** | 3.3177*** | 3.7641*** | 1.1785*** | 1.5053*** | 1.8504*** | 3.9171*** | 4.2003*** | 4.4720*** | 0.5082*** | 0.8954*** | 1.1999*** |
| | (0.024) | (0.022) | (0.038) | (0.035) | (0.034) | (0.040) | (0.023) | (0.022) | (0.019) | (0.035) | (0.034) | (0.034) |
| Region FE | Yes | Yes | Yes | Yes | Yes | Yes | Yes | Yes | Yes | Yes | Yes | Yes |
| Year FE | Yes | Yes | Yes | Yes | Yes | Yes | Yes | Yes | Yes | Yes | Yes | Yes |
| Obs. | 5,877 | 5,877 | 5,877 | 3,392 | 3,392 | 3,392 | 5,935 | 5,935 | 5,935 | 5,974 | 5,974 | 5,974 |

Standard errors in parentheses

*** p<0.01,

** p<0.05,

* p<0.1

transportation issues for lower quartile counties should be an area of focus for policy makers, but can be less of a focus for counties with fewer low access individuals.

### 3.4 RQ4—How does the relationship differ across storms with a local versus broad impact?

Previous estimations considered the count of all extreme weather events regardless of type. However, it is also important to consider how individual storm types may be associated with access and availability. We estimate the log-linear specification with region and year fixed effects for storms that are more likely to have localized effects, i.e. tornados, severe thunderstorms, extreme wind events, and floods, and for events that have more broad geographic impacts, i.e. winter weather storms, hurricanes and tropical storms, and wildfires. Table 5 presents the results of these event type estimations.

The relationship between localized events and availability generally follow the previous specifications with an extreme weather event being associated with a lower store density. The association was the strongest for grocery stores and the weakest for convenience stores. For example, an additional severe thunderstorm is associated with a 1.27 percent reduction in grocery store density, but is only associated with a 0.32 percent reduction in convenience store density. In contrast, the relationship between localized events and accessibility is less straightforward. An additional tornado or extreme wind event is associated with a 2.30 percent and a 0.56 percent lower proportion of the population with low access, respectively, while an

**Table 5. Estimation results for the relationship between localized and broad event types and access and availability.**

| VARIABLES | More Localized Events | | | | Broad Geographic Events | | | |
|---|---|---|---|---|---|---|---|---|
| | Grocery | Supercenter | Convenience | Low-Access | Grocery | Supercenter | Convenience | Low-Access |
| Number of Tornados | -0.0163*** | -0.0051 | -0.0107** | -0.0230*** | | | | |
| | (0.006) | (0.007) | (0.004) | (0.007) | | | | |
| Number of Severe Thunderstorms | -0.0127*** | -0.0051*** | -0.0032*** | 0.0079*** | | | | |
| | (0.002) | (0.002) | (0.001) | (0.002) | | | | |
| Number of Extreme Wind Events | 0.0022 | -0.0200*** | -0.0094*** | -0.0056** | | | | |
| | (0.002) | (0.002) | (0.001) | (0.003) | | | | |
| Number of Floods | -0.0165*** | -0.0107*** | -0.0092*** | 0.0123*** | | | | |
| | (0.003) | (0.003) | (0.002) | (0.003) | | | | |
| Number of Winter Weather Storms | | | | | -0.0036 | 0 | 0.0044** | 0.0041 |
| | | | | | (0.003) | (0.004) | (0.002) | (0.006) |
| Number of Hurricane and Tropical Storms | | | | | 0.0169 | -0.0516*** | 0.0003 | 0.0536* |
| | | | | | (0.014) | (0.017) | (0.010) | (0.029) |
| Number of Wildfires | | | | | -0.0451*** | -0.0180* | -0.0220*** | -0.0038 |
| | | | | | (0.010) | (0.011) | (0.007) | (0.019) |
| Percent of population non-Hispanic Black | 0.0034*** | -0.0044*** | 0.0031*** | 0.0091*** | 0.0017** | -0.0047*** | 0.0025*** | 0.0097*** |
| | (0.001) | (0.001) | (0.001) | (0.001) | (0.001) | (0.001) | (0.001) | (0.001) |
| Percent of population Hispanic | -0.0039*** | -0.0064*** | -0.0020*** | -0.0083*** | -0.0022*** | -0.0058*** | -0.0013** | -0.0085*** |
| | (0.001) | (0.001) | (0.001) | (0.001) | (0.001) | (0.001) | (0.001) | (0.001) |
| Percent of population Asian | 0.0035 | -0.0555*** | -0.0348*** | -0.0494*** | 0.0026 | -0.0591*** | -0.0376*** | -0.0506*** |
| | (0.004) | (0.004) | (0.003) | (0.005) | (0.004) | (0.005) | (0.003) | (0.005) |
| Percent of population AAPI | 0.0062*** | -0.0048 | 0.0006 | 0.0246*** | 0.0067*** | -0.0041 | 0.001 | 0.0244*** |
| | (0.002) | (0.003) | (0.001) | (0.002) | (0.002) | (0.003) | (0.001) | (0.002) |
| Metropolitan County | -0.3429*** | -0.4147*** | -0.3080*** | -0.3552*** | -0.3696*** | -0.4421*** | -0.3262*** | -0.3434*** |
| | (0.015) | (0.019) | (0.011) | (0.019) | (0.015) | (0.019) | (0.011) | (0.018) |
| Constant | 3.3012*** | 1.4266*** | 4.2172*** | 0.9837*** | 3.1995*** | 1.3400*** | 4.1597*** | 1.0303*** |
| | (0.018) | (0.026) | (0.013) | (0.022) | (0.016) | (0.023) | (0.011) | (0.020) |
| Region Fixed Effects | Yes | Yes | Yes | Yes | Yes | Yes | Yes | Yes |
| Year Fixed effects | Yes | Yes | Yes | Yes | Yes | Yes | Yes | Yes |
| Observations | 5,877 | 3,392 | 5,935 | 5,974 | 5,877 | 3,392 | 5,935 | 5,974 |
| R-squared | 0.178 | 0.365 | 0.262 | 0.237 | 0.168 | 0.349 | 0.255 | 0.236 |

Standard errors in parentheses

*** p<0.01,

** p<0.05,

* p<0.1

additional severe thunderstorm or flood event is associated with a 0.79 percent and 1.23 percent increase in the proportion, respectively. One reason for this difference may be that there is still variation within a single event type. For example, a flood event could happen in a relatively rural county next to a tributary or could be coastal flooding. In the first case, a flood zone may be associated with lower property development and population density, while in the second case development and population density, and therefore store density, would remain high.

There is a much weaker relationship between broad geographic extreme weather events and both availability and accessibility. Wildfires are the only event that are consistently associated with lower store densities across all store types. Hurricanes and tropical storms also have

a negative association, but only with supercenter density. Similarly, there is no statistically significant association between the broad geographic events and accessibility. This non-significance could be related to these events being more likely to be forecasted–and in turn may provide more time to anticipate impacts and prepare. Another potential reason could be that these storms are more geographically widespread, so stores would have to leave an entire region for this association to be discernable.

## 4. Discussion

In this research, we provide a first look into the relationship between the FRE and exposure to historical extreme events. Descriptively, we find that historical extreme weather event exposure and FRE measures have spatial overlap (RQ1). However, due to many confounding factors related to county characteristics, it is difficult to draw definitive conclusions using mapping alone. We, therefore, employ a modeling strategy where we include population, county, and regional controls. We find that FRE measures are related to extreme event exposure via event counts (RQ2), with higher counts of events related to lower food access and availability (e.g., fewer grocery stores, convenience stores and superstores). We also find that an increase in event counts is related to a higher proportion of the population that are low-income and low access with no vehicle. Such findings suggest there are indeed relationships between places with increased extreme events and their food environment–even after controlling for a variety of factors–warranting exploration in further detail.

We also examine the distribution of food access and availability (RQ3) and find heterogenous relationships, where extreme weather events have a lower proportional impact on the least-store density places (i.e., lower quartile), and higher for most-store density areas (i.e., upper quartile). The lower magnitude association in the lowest quartile does not necessarily mean that counties are well prepared for extreme weather events, but motivates different approaches from a policy perspective. We also find heterogenous relationships across the distribution of the proportion of the population that is low-income, low access and does not own a vehicle. For those areas with the lowest accessibility (lowest quartile), extreme weather is associated with a larger proportion of the population being low-income, low access and no vehicle. For areas with greater accessibility (all other quartiles) the relationship was insignificant.

Finally, we consider how different storm event types may be related to access and availability (RQ4). We find that for event types that typically are more localized in nature, such as thunderstorms or flooding, there is an association with event counts and proportion of the population that is low access. However, for broader impacting events, such as hurricanes and tropical storms, the relationship is less robust, with only superstore density being related to these event counts. We offer two potential explanations for this lack of association compared to more localized events. First, these types of events are often easily forecasted, so there could be more time to prepare. Second, the storms are widespread and seasonal, so stores would have to leave entire regions in order to avoid the storms. For both these reasons, stores and individuals may be less likely to make major adjustments in location compared to more random localized events.

While we find evidence that various dimensions of the food environment, such as access and availability, are related to extreme event exposure, we are careful not to draw any causal conclusions about these relationships. However, the existence of these associations suggest, at minimum, that aspects of the food environment should be explored within the context of extreme weather events because extreme weather may exacerbate already existing food access issues. For example, places with low food access have a greater dependence on smaller privately

owned grocery and convenience stores that may not have business continuity plans for natural disasters like those required of larger national chains [67]. In addition, smaller convenience stores often carry fewer fresh fruits and vegetables and more self-stable goods. Although, these foods found in convenience stores may "survive" better in an extreme weather event, they may also exacerbate already existing health disparities because areas that are heavily reliant on convenience stores are also disproportionately communities of color [68].

There are opportunities for policy makers to implement programs that generate multiple, cross-cutting benefits to particularly vulnerable communities. For example, Federal, state, and local programs like the Healthy Food Finance Initiative and Philadelphia Fresh Food Financing Initiative can both increase access to healthy food, decrease food insecurity, and increase community resilience to extreme weather events [69]. Prior literature has noted that building a more robust food system benefits the local economy through job creation and increased supply chain strength for both food and non-food systems [70], while addressing each aspect of the Global Syndemic, including the climate crisis, obesity, and malnutrition. Similarly, a more robust food system can mitigate the immediate impacts of an extreme weather event and allow for a quicker and stronger long-term recovery.

Past literature has noted specific policy recommendations, centered around enhancing infrastructure in food processing and distribution, that would lead to a more resilient food system. These recommendations include bolstering road networks, electrical power systems, fuel supply transportation systems, and storage and distribution infrastructure [71]. For example, most food is transported by truck, therefore building redundant transportation networks that do not include vulnerable structures such as bridges and tunnels helps ensure resilience in the face of closures. Locating critical warehouse suppliers in non-high-risk areas can also ensure a safe adequate supply of food during extreme weather.

In addition to physical improvements, city and statewide disaster relief planners should coordinate with local governments, NGOs, and food retailers to conduct food system resilience assessments that take into account the local risks and unique vulnerabilities of the area. In particular, plans should prioritize and include provisions for neighborhoods with low food access and high disaster impact risks [67]. Although policies can incentivize physical improvements on the production side, these disaster relief plans should consider limitations, e.g. refrigeration, cooking facilities, etc., on the consumer side. These are particularly important when recovering from an event, but can also support healthy eating in general. Our results support these policy recommendations by drawing an initial relationship between high extreme weather exposure and limited FRE access and availability.

Although this paper establishes a relationship between common FRE measures and exposure to extreme weather events, there are some limitations to consider when interpreting the results. First, the level of analysis is on the county level geographically and on the year level temporally. Previous literature has shown that administrative boundaries, such as county boundaries, may not accurately capture nuances in the FRE [72]. For example, all stores in a county may be located in a particular region, which a county level density will not be able to account for. Similarly, having a finer temporal level, e.g. monthly or weekly, would allow us to consider seasonality of extreme weather. However, the impacts of these two data limitations are likely minor for two main reasons. First, our aim is to establish a high-level relationship between extreme weather exposure and low food access and availability, rather than to establish determinants at a household or small geographic level. Second, these two limitations would likely introduce classical measurement error into the data. Classical measurement error leads to attenuation bias, or a bias towards zero in estimates. Therefore, if classical measurement error is present, the estimates can be considered lower bounds and the statistical significance is still valid.

## 5. Conclusion

Our research provides a systematic exploration into the relationship between food energy environments and extreme weather exposure. In our analysis of the data, we found evidence that suggests that the same areas that have higher exposure to extreme weather also tend to have comparatively lower FRE access and availability. While this research is unable to disentangle the underlying or causal factors for why this may be the case, it motivates further research. The benefits of more fully understanding the connection between food security and uncontrollable extreme weather events can extend beyond policy to mitigate their impact. For example, public safety power shutoffs, i.e. a pre-emptive de-energization of power infrastructure intended to prevent wildfires, can cause loss of power to wide geographic areas and can have similar impacts as extreme weather events in the absence of the event itself (wildfire). In this respect, insights into the relationships we have identified in this research can also help inform policy meant to mitigate the impacts of other disruptive events, that are not necessarily weather-related in origin, but could have similar effects. To this end, future research considering topics in this area should strive to utilize causal research designs on smaller scales in order to identify how disruptive events, weather-related or otherwise, may be shaping the food environment, changing retailer and consumer decisions, and what policy instruments promote systems that are robust to these shocks.

## Supporting information

**S1 Data.**
(DTA)

## Author Contributions

**Conceptualization:** Benjamin Scharadin, Chad Zanocco.

**Data curation:** Benjamin Scharadin, Chad Zanocco, Jacqueline Chistolini.

**Formal analysis:** Benjamin Scharadin.

**Methodology:** Benjamin Scharadin, Chad Zanocco.

**Project administration:** Benjamin Scharadin, Chad Zanocco.

**Visualization:** Benjamin Scharadin, Chad Zanocco, Jacqueline Chistolini.

**Writing – original draft:** Benjamin Scharadin, Chad Zanocco, Jacqueline Chistolini.

**Writing – review & editing:** Benjamin Scharadin, Chad Zanocco, Jacqueline Chistolini.

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
