## [Decision Letter · Decision Letter 0]

28 Apr 2023

PONE-D-23-04647Food retail environments, extreme weather, and their overlap: exploratory analysis and recommendations for US food policyPLOS ONE

Dear Dr. Scharadin,

Thank you for submitting your manuscript to PLOS ONE. After careful consideration, we feel that it has merit but does not fully meet PLOS ONE’s publication criteria as it currently stands. Therefore, we invite you to submit a revised version of the manuscript that addresses the points raised during the review process.

We look forward to receiving your revised manuscript.

Kind regards,

Larissa Loures Mendes, Ph.D.

Academic Editor

PLOS ONE

Journal Requirements:

3. We note that Figure 1 in your submission contain map image which may be copyrighted. All PLOS content is published under the Creative Commons Attribution License (CC BY 4.0), which means that the manuscript, images, and Supporting Information files will be freely available online, and any third party is permitted to access, download, copy, distribute, and use these materials in any way, even commercially, with proper attribution. For these reasons, we cannot publish previously copyrighted maps or satellite images created using proprietary data, such as Google software (Google Maps, Street View, and Earth). For more information, see our copyright guidelines: http://journals.plos.org/plosone/s/licenses-and-copyright.

Additional Editor Comments:

Dear Authors,

I appreciate the opportunity to review this interesting manuscript that assesses the relationship between the occurrence of extreme weather events and population with limiting food retail environments. I believe that after incorporating the reviewers' suggestions, the manuscript will be even more robust for publication.

Reviewers' comments:

Reviewer's Responses to Questions

**Comments to the Author**

1. Is the manuscript technically sound, and do the data support the conclusions?

Reviewer #1: Yes

Reviewer #2: Yes

2. Has the statistical analysis been performed appropriately and rigorously? 

Reviewer #1: Yes

Reviewer #2: Yes

3. Have the authors made all data underlying the findings in their manuscript fully available?

Reviewer #1: No

Reviewer #2: Yes

4. Is the manuscript presented in an intelligible fashion and written in standard English?

Reviewer #1: Yes

Reviewer #2: Yes

5. Review Comments to the Author

Reviewer #1: This is an interesting manuscript that studies the relationship between the occurrence of extreme weather events and population with limiting food retail environments. However, I have some suggestions that are important to authors consider to include e modifications in the text. It's important to think that this manuscript will be read by government managers. Therefore, adapting the tone of the text and specifying the suggestions is crucial.

Introduction

The introduction is interesting but too long. The topics of food environments and extreme weather events seem to be part of your methods, think about it. Leave only the topic background and motivation as the introduction. This topic explains well the motivation of the study and the gap it will fill. And you can add the research questions at the end of your introduction.

Methods

The information about the data source should be before the regression topic. We need to understand first what kind of data you have so that we can understand the analyses.

You mentioned food deserts in the introduction. Why don't you include this analysis in your study?

Discussion

I miss some discussions about the availability of healthy food and food security. Both are mentioned briefly in the text. We need to discuss what kind of food "survives" better to these extreme events. For example, convenience stores sell more unhealthy food. So, are they more present in areas impacted by extreme weather events?

Health and fresh food are more affected by these events, because of the production type, right?

And all of these scenarios affect food security, but you mentioned it only in the conclusion.

Reviewer #2: This is an interesting study the explores the relationship between geographic areas with higher exposure to extreme weather events and areas that have lower access and availability to food retailers.

Please suggest that both in the introduction and the discussion, it is important to analyze and discuss the results from the perspective of the global syndemic (https://pubmed.ncbi.nlm.nih.gov/30700377/). The three pandemics—obesity, undernutrition, and climate change—represent The Global Syndemic that affects most people in every country and region worldwide. They constitute a syndemic, or synergy of epidemics, because they co-occur in time and place, interact with each other to produce complex sequelae, and share common underlying societal drivers (Swinburn et al., 2019).

Line 82 [1.2 Food Environments] -> Please include the dimensions of the food environment as proposed by Glanz et al. (2005)

Glanz, K., James, F., Sallis, B. E. S., & Lawrence, D. F. (2005). Healthy nutrition environments: Concepts and measures. American Journal of Health Promotion, 19(330–333), ii.

- It would be important to include measures that emphasize access to healthy facilities. Did the authors consider using the food desert methodology as a measure of the food environment? Food deserts are often characterized as socioeconomically vulnerable neighborhoods, where individuals have poor access to healthy food. It is important to emphasize that the component ‘desert’ in this term is inherently spatial and relates to physical lack of food outlets providing healthy food options in low income neighborhoods. In contrast, food swamps are neighborhoods that have large numbers of unhealthy food outlets, wherein strong marketing strategies constantly target and promote this food type. The Modified Food Retail Environment Index considers supermarkets, hypermarkets and fruit stores as healthy food outlets; fast-food restaurants and convenience stores are considered unhealthy food outlets.

Line 199 – Please, make clear the geographic unit of the four food retail environment measures.

[DISCUSSION] – Please, when discussing recommendations for public policies, it is important to discuss policies that aim to contribute to healthy and sustainable food systems and that improve access to healthy food, especially among more vulnerable populations.

6. PLOS authors have the option to publish the peer review history of their article (what does this mean?). If published, this will include your full peer review and any attached files.

Reviewer #1: No

Reviewer #2: No

---

## [Author Response · Author response to Decision Letter 0]

15 Jun 2023

Response to reviewers for “Food retail environments, extreme weather, and their overlap: exploratory analysis and recommendations for US food policy”

Reviewer feedback is in bold text | Response to reviewer is in plain text

Reviewer #1: 

This is an interesting manuscript that studies the relationship between the occurrence of extreme weather events and population with limiting food retail environments. However, I have some suggestions that are important to authors consider to include e modifications in the text.

• It's important to think that this manuscript will be read by government managers. Therefore, adapting the tone of the text and specifying the suggestions is crucial.

Response: We agree that this manuscript will likely be read by government managers and others in the policy space and agree that the tone, in turn, needs to match that. In response, we have attempted to present the analysis and findings in a way that is accurate and descriptive without using unnecessary jargon. This includes minor edits where reasonable throughout the manuscript to adapt a tone for this audience in mind. However, it is also important that the manuscript accurately communicates our analysis and findings, and have ensured that any changes to the text do not have negative impacts on these aspects.

We also agree that providing additional specificity in our connection to policy recommendations would improve the manuscript and make it more useful for a policy audience. Accordingly, we made updates throughout the revised manuscript, some which we have provided below:

1. Discuss how programs already in place, like the Healthy Food Finance Initiative and Philadelphia Fresh Food Financing, can address multiple dimensions of this issue. 

2. Make specific suggestions that future policies should include, such as constructing warehouse facilities in low event areas.

3. Provide broader considerations, such as acknowledging the tension between the “survival” of shelf stable goods and their lower nutritional quality.

Through including more detailed policy suggestions, we believe we provide a level specificity that is useful for policy makers, while still allowing the suggestions to be flexible to the intricacies of unique localities.

Introduction

• The introduction is interesting but too long. The topics of food environments and extreme weather events seem to be part of your methods, think about it.

Response: Thank you for suggesting the reorganization of some of the information in the introduction section. We agree that the food environment section and the extreme weather section are more appropriate in the methods section. In response to your comment we now include the discussion of how past literature has characterized each directly before discussing our data sources and variable construction in the revised manuscript in lines 99 through 165 for the food environment and lines 166 through 264 for extreme weather.

This revised organization better allows the reader to follow how past literature informed our variable construction and has significantly shortened the introduction section. In addition, we have also edited the introduction content that was not moved to other sections of the manuscript to introduce the topic more efficiently.

• Leave only the topic background and motivation as the introduction. This topic explains well the motivation of the study and the gap it will fill. You can add the research questions at the end of your introduction.

o Response: Thank you again for the detailed suggestion and noting that the introduction provides good motivation for the study and how it will address a current gap in literature. We have taken your advice and now have only two sections in the introduction in the revised manuscript: the background and motivations section and the research questions section.

Methods

• The information about the data source should be before the regression topic. We need to understand first what kind of data you have so that we can understand the analyses.

Response: Thank you for this suggestion. In response, we have moved the data section to be first in the Materials and Methods section in the revised manuscript. As noted above, this is now combined with our variable construction discussion that was in the introduction in the original manuscript. We agree that this structure makes it easier for the reader to understand data and variable construction fully prior to the regression specification.

• You mentioned food deserts in the introduction. Why don't you include this analysis in your study?

Response: Thank you for this note. Although we originally only include the term food desert in the introduction of the original manuscript, we believe we do conduct analysis on the relationship between extreme weather events and food deserts because the three food environment measures we use are components in defining the term food desert. We have now clarified this in the manuscript and provide some additional information below.

The USDA defines food deserts as census tracts that meet both low-income and low-access criteria including. These criteria are 1) a poverty rate greater than or equal to 20 percent or median family income not exceeding 80 percent statewide (rural/urban) or metro-area (urban) median family income, 2) at least 500 people or 33 percent of the population located more than 1 mile (urban) or 10 miles (rural) from the nearest supermarket or large grocery store. (Dutko, Ver Ploeg, and Farrigan 2012).

Instead of adapting this definition for use at the county level, which would be required because the extreme weather counts are at the county level, we use a continuous household variable that measures similar dimensions. We measure accessibility as the percentage of the population in a county that is low-income, has low-access to grocery stores, and does not have access to a vehicle. This is a continuous version of the tract level food desert definition, where instead of saying, “yes” a county or a tract has a value over this threshold in these measures, we simply keep the measure as a continuous variable.

In addition, we use two availability measures that provide more nuance than just the food desert definition. In this way, we are able to investigate the relationship between extreme weather events and the availability of healthy food retailers through the concentration of grocery stores and superstores and the availability of less healthy food retailers through the concentration of convenience stores.

In response to your comment, we have clarified the connection between the measures we use in our analysis and how food deserts are defined. Specifically, we have added clarifying statements on pages 7 through 9 in the revised manuscript. Thank you again for the note because it led to edits that clarified the revised manuscript.

Discussion

• I miss some discussions about the availability of healthy food and food security. Both are mentioned briefly in the text. We need to discuss what kind of food "survives" better to these extreme events. For example, convenience stores sell more unhealthy food. So, are they more present in areas impacted by extreme weather events?

Response: We agree that it is important to note what types of food “survive” events and the healthfulness of that food. We have added some discussion of this topic in lines 530 to 543 of the revised manuscript.

There is tension between food being shelf-stable and food being suggested for increased consumption, e.g. fruits and vegetables. As you note, convenience stores tend to carry more unhealthy, shelf-stable food, but this is also more likely to stay edible if there is no refrigeration or cooking facilities available. This has the potential to exacerbate already existing health disparities because areas that rely on convenience stores for food access are more likely to be communities of color.

In response to your feedback and the feedback of Reviewer #2, in the revised manuscript we now suggest that policies such as the Healthy Food Finance Initiative and Philadelphia Fresh Food Financing Initiative continue to be funded because they have the ability to address both food insecurity and resilience to extreme weather events.

• Health and fresh food are more affected by these events, because of the production type, right?

Response: Healthy and fresh food are more impacted by these types of events because of the perishability. This contributes to aspects of the production process because there is less time from harvest until the food can be consumed and because of shipping requirements, such as refrigeration. There are also limitations on the consumer side, where they would need access to refrigeration and cooking facilities. Many of these requirements may not be met during an extreme weather event.

In the revised manuscript, we have included a discussion of the perishability problem and have made suggestions that future planning consider both the production side limitations and consumer side constraints that occur because of extreme weather events.

• And all of these scenarios affect food security, but you mentioned it only in the conclusion.

Response: We agree that all of these scenarios impact food security. The four measures we use in our study are components of food insecurity. As a result, when we mention how availability and accessibility are related to and impacted by extreme weather events, we are talking about how those impact food insecurity. However, we have made it more clear in the discussion by specifically mentioning food insecurity throughout the revised manuscript. One example occurs in the lines 535 to 543 in the revised manuscript.

Reviewer #2: 

This is an interesting study the explores the relationship between geographic areas with higher exposure to extreme weather events and areas that have lower access and availability to food retailers.

• Please suggest that both in the introduction and the discussion, it is important to analyze and discuss the results from the perspective of the global syndemic (https://pubmed.ncbi.nlm.nih.gov/30700377/). The three pandemics—obesity, undernutrition, and climate change—represent The Global Syndemic that affects most people in every country and region worldwide. They constitute a syndemic, or synergy of epidemics, because they co-occur in time and place, interact with each other to produce complex sequelae, and share common underlying societal drivers (Swinburn et al., 2019).

Response: Thank you for suggesting that we introduce the Global Syndemic as a framework to motivate our research question. We agree that it is an important framework to consider when analyzing the relationship between food insecurity measures and extreme weather events. In response to your comment, we have included an additional paragraph in the introduction, discussing how the Global Syndemic is present throughout the world and the U.S. and motivates the need to understand how seemingly unrelated topics, such as climate change and grocery store density are related. The new introduction paragraph is on page 2 in the revised manuscript.

We also consider the Global Syndemic in the discussion section on page 32 in the revised manuscript. In the original manuscript we discussed how policies have the opportunity to address both everyday food insecurity issues, while also making vulnerable communities more robust to extreme weather shocks. Hence the policies should co-occur in time and place in the same way that the pandemics are co-occuring in time and place.

• Line 82 [1.2 Food Environments] -> Please include the dimensions of the food environment as proposed by Glanz et al. (2005)

Glanz, K., James, F., Sallis, B. E. S., & Lawrence, D. F. (2005). Healthy nutrition environments: Concepts and measures. American Journal of Health Promotion, 19(330–333), ii.

o Response: Thank you for noting the contribution of Glanz et al. (2005) to the conceptual food retail environment literature. We have cited the work in line 107 on page 6 of the revised manuscript when discussing dimensions of the food retail environment.

• It would be important to include measures that emphasize access to healthy facilities.

o Response: Thank you for this note and we agree that distinguishing between healthy and less healthy food retailers is important. We use three availability measures that provide more nuance than a single store density measure, in part to note healthy and less healthy. In this way, we are able to investigate the relationship between extreme weather events and the availability of healthy food retailers through the concentration of grocery stores and superstores and the availability of less healthy food retailers through the concentration of convenience stores. We have made clarifying edits throughout the paper to emphasize that convenience stores can be considered less healthy than grocery stores and supercenters.

• Did the authors consider using the food desert methodology as a measure of the food environment? Food deserts are often characterized as socioeconomically vulnerable neighborhoods, where individuals have poor access to healthy food. It is important to emphasize that the component ‘desert’ in this term is inherently spatial and relates to physical lack of food outlets providing healthy food options in low income neighborhoods. In contrast, food swamps are neighborhoods that have large numbers of unhealthy food outlets, wherein strong marketing strategies constantly target and promote this food type. The Modified Food Retail Environment Index considers supermarkets, hypermarkets and fruit stores as healthy food outlets; fast-food restaurants and convenience stores are considered unhealthy food outlets.

Response: Thank you for this note. We believe we do conduct analysis on the relationship between extreme weather events and food deserts because the three food environment measures we use are components in defining a food retail environment and thus if below a certain threshold in defining a food desert. Reviewer 1 had a similar comment (see Reviewer #1’s second comment under “Methods”) which we responded to in detail with more information. In response to your feedback, and the feedback of Reviewer #1, we have clarified the connection between the measures we use in our analysis and how food deserts are defined. Specifically, we have added clarifying statements on pages 7 through 9 in the revised manuscript. Thank you again for the note because it led to edits that clarified the revised manuscript.

• Line 199 – Please, make clear the geographic unit of the four food retail environment measures.

o Response: Thank you for this note. We agree that clearly stating the geographic unit is important. We have clarified that the measures are on the county level in line 144 of the revised manuscript.

• [DISCUSSION] – Please, when discussing recommendations for public policies, it is important to discuss policies that aim to contribute to healthy and sustainable food systems and that improve access to healthy food, especially among more vulnerable populations.

o Response: Thank you for this suggestion, which is similar to comments from Reviewer #1, that we provide more specific information in our discussion about policy recommendations. We agree that it is important for policy recommendations to aim to contribute to healthy and sustainable food systems. In the discussion section of the revised manuscript, we now discuss policies that will help increase access to healthy food, such as the Healthy Food Finance Initiative and Philadelphia Fresh Food Financing Initiative, which can both increase access to healthy food, decrease food insecurity, and increase community resilience to extreme weather events. In particular, we mention that these policies do and should continue to particularly focus on BIPOC communities in the future, where reliance on convenience stores is higher.

---

## [Decision Letter · Decision Letter 1]

17 Jul 2023

Food retail environments, extreme weather, and their overlap: exploratory analysis and recommendations for US food policy

PONE-D-23-04647R1

Dear Dr. Scharadin,

We’re pleased to inform you that your manuscript has been judged scientifically suitable for publication and will be formally accepted for publication once it meets all outstanding technical requirements.

Kind regards,

Larissa Loures Mendes, Ph.D.

Academic Editor

PLOS ONE

Dear Authors,

I am grateful for the revisions made, I believe that the study was even more interesting for publication.

Yours sincerely

Reviewers' comments:

Reviewer's Responses to Questions

**Comments to the Author**

1. If the authors have adequately addressed your comments raised in a previous round of review and you feel that this manuscript is now acceptable for publication, you may indicate that here to bypass the “Comments to the Author” section, enter your conflict of interest statement in the “Confidential to Editor” section, and submit your "Accept" recommendation.

Reviewer #1: All comments have been addressed

Reviewer #2: All comments have been addressed

2. Is the manuscript technically sound, and do the data support the conclusions?

Reviewer #1: Yes

Reviewer #2: Yes

3. Has the statistical analysis been performed appropriately and rigorously? 

Reviewer #1: Yes

Reviewer #2: Yes

4. Have the authors made all data underlying the findings in their manuscript fully available?

Reviewer #1: Yes

Reviewer #2: Yes

5. Is the manuscript presented in an intelligible fashion and written in standard English?

Reviewer #1: Yes

Reviewer #2: Yes

6. Review Comments to the Author

Reviewer #1: (No Response)

Reviewer #2: The authors have adequately addressed mine comments. The manuscript is acceptable for publication. I recommend the manuscript for publication.

7. PLOS authors have the option to publish the peer review history of their article (what does this mean?). If published, this will include your full peer review and any attached files.

Reviewer #1: **Yes: **Luana Lara Rocha

Reviewer #2: No

---

## [Editor Report · Acceptance letter]

19 Jul 2023

PONE-D-23-04647R1 

Food retail environments, extreme weather, and their overlap: exploratory analysis and recommendations for U.S. food policy 

Dear Dr. Scharadin:

I'm pleased to inform you that your manuscript has been deemed suitable for publication in PLOS ONE. Congratulations! Your manuscript is now with our production department. 

Kind regards, 

on behalf of

Dr. Larissa Loures Mendes 

Academic Editor

PLOS ONE